# Views of prison staff in Scotland on the potential benefits and risks of e-cigarettes in smoke-free prisons: a qualitative focus group study

Ashley Brown,[1] Helen Sweeting,[2] Sean Semple,[1] Linda Bauld,[3] Evangelia Demou,[2] Greig Logan,[4] Kate Hunt[1]

[1]Institute for Social Marketing and Health, University of Stirling Institute for Social Marketing, Stirling, UK
[2]MRC/CSO Social & Public Health Sciences Unit, University of Glasgow, Glasgow, UK
[3]Usher Institute of Population Health Sciences and Informatics, University of Edinburgh, Edinburgh, UK
[4]Institute of Health and Wellbeing, University of Glasgow, Glasgow, UK

**Correspondence to**
Ms Ashley Brown;
a.l.brown@stir.ac.uk

## ABSTRACT

**Objective** Electronic cigarettes (e-cigarettes) were introduced into all Scottish prisons in February 2018, some months after prisons began preparing in 2017 for a smoking ban implemented in November 2018. In 2016/2017, prison staff views on the potential benefits and risks of e-cigarettes were explored in advance of the introduction of: (1) a smoking ban and (2) e-cigarettes.

**Setting** Fourteen prisons in Scotland.

**Participants** Seventeen focus groups and two paired interviews were conducted with 132 staff in 14 Scottish prisons 4–9 months before plans for a smoking ban were announced in July 2017. Both smoking and non-smoking staff were invited to participate.

**Results** Prison staff highlighted three potential risks of e-cigarettes in smoke-free prisons: staff health risks from e-cigarette vapour; prisoner health risks from vaping; and risks to both groups from e-cigarette misuse, defects or accidents. Conversely, potential benefits of e-cigarettes in smoke-free prisons centred on: reducing smoking-related health harms to staff and prisoners; helping prisoners to manage without tobacco; and supporting staff to maintain safety and discipline in prison. Staff who participated in focus groups had limited experience of vaping and expressed some uncertainty and misunderstandings about e-cigarettes.

**Conclusion** Our findings highlight that scientific uncertainty, misunderstanding about vaping, the complexity of prisons as workplaces and prison tobacco control policy all have implications for staff perceptions of the potential place of e-cigarettes in smoke-free prisons. To alleviate staff concerns, there is a need for reliable information on e-cigarettes. Staff may also require reassurances on whether products are 'tamper proof', and rules about vaping indoors.

## Strengths and limitations of this study

► To our knowledge, this is the first study to examine staff views on the potential benefits and risks of introducing e-cigarettes for prisoners in a smoke-free prison system, using data collected prior to the announcement of a smoking ban and before there was experience of e-cigarette use in prisons.

► This study provides valuable insight into the potential place of e-cigarettes in a complex workplace setting in which staff were, on the one hand, still exposed to secondhand smoke, while on the other increasingly aware that they could be responsible for the management and care of vulnerable and/or challenging individuals subject to enforced smoking abstinence.

► Our results are based on analysis of rich qualitative data collected from a relatively large sample of prison staff from 14 Scottish prisons, with varied smoking histories and experience of working with diverse prisoner groups.

► It is important to acknowledge that: the sample was self-selecting, few members of staff had direct experience of vaping and staff views may have developed since these qualitative data were collected.

for combustible tobacco cigarettes reduces users' exposures to numerous toxins and carcinogens present in combustible tobacco cigarettes.' A similar view on e-cigarettes has been reached by UK health organisations, including, for example, the Royal College of Physicians,[2] Public Health England[3 4] and NHS Health Scotland.[5] There is however greater disagreement within academic and health communities over issues such as the effectiveness of e-cigarettes as a smoking cessation aid[6] and the potential influence of e-cigarettes on smoking norms and uptake among adults and young people.[1 7]

The contested nature of e-cigarettes is perhaps unsurprising. The introduction of a new technology is inevitably accompanied by

## INTRODUCTION

There is a growing consensus that using e-cigarettes (vaping) is safer than smoking conventional cigarettes. A review of the scientific evidence on e-cigarettes, conducted by the National Academies of Sciences, Engineering and Medicine,[1] reported 'conclusive evidence that completely substituting e-cigarettes

a period of scientific uncertainty and debate about how to regulate technology in the absence of conclusive evidence about its health effects. The 'precautionary principle' may be relevant in situations of uncertainty. The WHO's definition of the principle centres on the idea that '… scientific uncertainty should not be used as a reason to postpone preventive measures'.[8] The precautionary principle requires complex judgements (even conjectures) about the likely balance of potential benefits and risks of introducing different types of preventive measures versus not implementing these measures.[9] With respect to e-cigarettes, Fairchild and Bayer[10] suggest that beliefs among some experts that e-cigarettes are a potential threat to health and require strict regulation are 'shaped implicitly by a precautionary impulse'. On the other hand, it has been suggested that experts who believe that e-cigarettes are part of the solution to the challenge of tobacco control might be influenced by the principle of harm reduction.[10] Harm reduction strategies are informed by a belief that steps should be taken to minimise harm from tobacco and other drug use in circumstances in which abstinence is not achievable.[11]

In the UK, it is illegal to smoke in most enclosed workplaces and public spaces. As both residential settings and workplaces, prisons have historically adopted a distinctive approach to tobacco control, including partial exemption from smoke-free laws in the UK.[12] However, ongoing concerns about Secondhand Smoke (SHS) exposures are one factor in recent decisions by the UK and Scottish governments to extend smoke-free policies to all indoor and outdoor areas of prisons in Wales and England[13] (rolling out from 2016) and Scotland (from 30 November 2018 in all 15 prisons).[14]

E-cigarettes have been available for prisoners to buy in some English and Welsh prisons from 2016 and were first made available for prisoners to buy in all Scottish prisons in early 2018 (~10–15 months after the data presented in this paper were collected). The potential for e-cigarettes to help some prisoners to remain smoke-free is recognised by organisations such as NHS Health Scotland,[15] while other commentators have discussed the potential negative health and organisational effects of selling e-cigarettes in prisons which have implemented smoking bans.[16 17]

Commentary and research on e-cigarettes and their place in smoke-free environments has largely focused on the opinions of public health experts and the general public.[18 19] In-depth qualitative research examining employees' views on vaping in particular settings is required to help with the development of acceptable and effective workplace policies and measures on e-cigarettes. We believe the Tobacco In Prisons study (TIPs), as reported here, is one of the first studies to address specific evidence gaps in respect of e-cigarettes in one workplace, prisons, and to investigate views on the potential role of e-cigarettes in accompanying the removal of tobacco across a country's prison system. In this qualitative paper, we present prison staff views, using data from phase 1 of TIPs. These data were collected from 132 prison staff in 14 Scottish prisons several months before the July 2017 announcement that a comprehensive ban on smoking would be introduced from 30 November 2018 and prior to any significant policy debate in Scotland about the sale of e-cigarettes to prisoners. This paper extends previous reporting of prison staff and prisoner views on prison smoking bans, which only includes brief mention of the potential place of e-cigarettes in smoke-free prisons.[20] Here, we use the staff focus group data to explore in detail staff views on the specific benefits and risks of e-cigarettes. The research could help with the development of strategies in respect of e-cigarettes in prison and so support the successful introduction of smoke-free policies, and help reduce tobacco-related harms, not just in Scotland (where prisons have recently gone smoke-free) but in other jurisdictions that are considering implementing bans in the future.

## METHODS

Data were collected prior to the announcement of plans to implement a smoking ban in Scottish prisons. At the time of data collection, prisoners were allowed to smoke tobacco in cells and outdoor areas; staff could not smoke on prison premises. Nobody (staff, prisoners or visitors) was permitted to vape in Scottish prisons during the period in which these data were collected (November 2016–April 2017).

### Patient and public involvement

TIPs was designed to ensure that the views of prisoners and prison staff, as expressed to a research team who were independent of the prison service, could be heard at different stages of the process of moving towards smoke-free prisons in Scotland. At all stages of the study, a Research Advisory Group which included staff from various parts of the prison service and representation from unions representing prison staff, has given extensive feedback on the overall design of the study and on study materials (including topic guides for qualitative interviews/focus groups).

### Sampling and recruitment

As reported elsewhere,[20] 17 focus groups and two paired interviews (hereafter referred to collectively as 'focus groups') were conducted with staff from 14 Scottish prisons which had been recruited through a point of contact in each prison. The reason for carrying out paired interviews on two occasions was that other prison staff who were due to participate in the focus group were unable to attend at short notice. We asked the point of contact to invite around eight prison staff to participate in a focus group with other staff from the same prison. To enable the research to explore the diversity of views on smoking in prisons and prison smoking bans within and between establishments, it was explained that ideally we would like the focus groups to include both smoking and

non-smoking staff in a range of work roles. While we had limited control over how focus groups were assembled by the prison point of contact, most displayed some diversity with respect to staff smoking status. Across the sample of 132 staff, 78 had never smoked (NS), 30 were ex-smokers (ExS) and 11 currently used tobacco cigarettes (S). The smoking status for 13 participants is not known (NK). Eight staff reported having ever used an e-cigarette: five were currently vaping (V) and three were no longer vaping (ExV). Although it was not possible to record information on staff job roles consistently, the majority of those who took part were Scottish Prison Service staff, while some worked in the prison for other agencies, such as the National Health Service. Scottish Prison Service staff were a mix of residential, operational and instructor officers, managerial roles and administrative posts. The 14 prisons in which staff were working were varied with respect to: prisoner population (eg, sex, age, and sentence length), capacity, security status and prison architecture.

Quotations are included to illustrate key perspectives, indicating the prison code, focus group and smoking status of each speaker (eg, KA04 S=prison K, group A, participant 04, Smoker). Codes were randomly allocated to prisons by the research team specifically for this paper to protect anonymity.

### Data collection
Focus groups were chosen for this study as they are well suited to understanding the diversity of viewpoints on a subject and how opinions are shaped by varying personal, environmental and social factors. Focus groups (range 5–12 participants) and the two paired interviews were conducted between November 2016 and April 2017 by a member of the TIPs research team. They were carried out in a room in each prison chosen by the point of contact. The topic guide covered: smoking and exposures to SHS within prisons; smoking norms and prevalence within prison; the 'culture' of smoking within prisons; management of nicotine addiction (including e-cigarettes) in prisons and wider society; and opinions on rules on smoking. Specific areas for discussion on e-cigarettes included: whether staff had used e-cigarettes or knew others who used e-cigarettes; opinions about e-cigarettes in general; views on what might be good or bad about prisoners or staff vaping in prison; and opinions on any issues which might be raised by allowing vaping in prisons. The researchers formulated questions using their own words (often in response to issues raised by the groups), adjusted the order of topics as appropriate, prompted further discussion where relevant and invited staff to raise any points which they thought were pertinent.

### Data analysis and reporting
With written consent from participants, focus groups were audio recorded and transcribed verbatim. Transcripts were checked and de-identified prior to management of the data. TIPs researchers (KH, HS, ED and GL) who conducted the fieldwork developed a descriptive coding scheme to bring together data on similar topics in preparation for detailed analysis. This coding scheme was devised using a combination of inductive and deductive techniques. The task of coding transcripts was split between TIPs researchers. Due to the relatively large volume of qualitative data, summaries with digital links to the raw data for all content relating to e-cigarettes were subsequently produced by AB using the Framework function in NVivo software (QSR international). AB used the data summaries and raw data to conduct thematic analysis. The process involved identifying different dimensions of staff opinions on e-cigarettes, grouping together dimensions which were similar to create themes and subthemes and naming the themes and subthemes.[21] KH conducted independent analysis of the data, and other authors read a sample of the data to familiarise themselves. Emergent themes were discussed and revised until an interpretation was agreed on by all authors. This paper largely follows Standards for Reporting Qualitative Research (SRQR) guidelines (see online supplementary file 1).

## RESULTS
### Background: personal experience and expressed knowledge of e-cigarettes
Most staff who participated in the focus groups had little personal experience of e-cigarettes; only a small number (n=8/132) reported having ever tried vaping. Consequently, a recurring theme in the focus groups was staff reporting low levels of knowledge about e-cigarettes, contributing to a sense of uncertainty and confusion about vaping. One way in which this emerged was in the different names which staff gave to e-cigarettes, including 'vaping machines' (JA05 NS), 'vapour sticks' (FA06 NS) and 'vape cigarette' (HA03 ExS). Another way in which uncertainty about e-cigarettes emerged was in the frequent questions which staff asked each other or the interviewer during discussions:

HA05 NS-Isn't it quite expensive as well e-cigarettes, are they not quite expensive?

HA02 NS-I don't know, I think they're getting cheaper, they were quite expensive.

Staff also seemed to be unsure about how e-cigarettes work and whether there is more than one type of product. Staff occasionally muddled 'tobacco' and 'nicotine' when discussing e-cigarettes, although mistakes in terminology were generally corrected by another member of the group. There were also examples where staff appeared to confuse e-cigarettes with a nicotine inhaler.

Uncertainty was also expressed by some staff about what is known about the health risks and safety of e-cigarettes:

FA01 NS: [w]ell I was just gonna ask, why are they banning vaping [in other contexts outside of prisons], is there something wrong with vaping? Because they're banning it in lots of different places...for some reason

In other cases, staff expressed greater awareness that there are gaps in the evidence on e-cigarettes, which was another source of uncertainty. It is interesting to note however that some staff seemed unfamiliar with the pace at which knowledge has accumulated in recent years:

BA10 NS: The last time they reviewed the policy, there was still no reliable research on the health implications of e-cigarettes.

Apparent misunderstandings about e-cigarettes were also identified. These included a staff member believing that it might be more dangerous for non-smokers to take up e-cigarettes compared with conventional smoking and statements made as 'facts', such as that e-cigarettes 'emit ten carcinogens as opposed to 100 from ordinary cigarettes' (CD27 ExS).

Hence, it was against this background that staff were evaluating the potential benefits and risks of vaping in prison, in the event that tobacco was removed from the prisons at some future date. Overall, opinions around allowing e-cigarettes in smoke-free prisons included views which could be described as positive, and others which were highly negative, with some staff reporting that they did not feel sufficiently well informed to have an opinion of any sort. Although only eight participants had experience of vaping, those with direct experience of e-cigarettes generally acknowledged their potential benefits in smoke-free prisons. However, there were exceptions; for instance, less positive views were expressed by a staff member who had themselves stopped vaping due to concerns about potential adverse health effects of e-cigarette vapour.

The range of potential risks and benefits of e-cigarettes in smoke-free prisons as discussed by all participants in the focus groups are explored in detail below.

### Perceptions of potential risks of e-cigarettes in a smoke-free prison

Potential risks of e-cigarettes in a smoke-free prison centred on three subthemes: potential risks to staff health from secondhand vapour; potential risks to prisoner health from vaping; and potential risks to staff and prisoners from device misuse, product defects or accidents.

#### Potential risks to staff health from secondhand vapour

Some non-smoking and smoking staff worried that they might be harmed by breathing the vapour from prisoners' e-cigarettes when they came to work. Several raised concerns that health risks from secondhand vapour exposures might be uncovered in the future:

LA8 ExS: …not everybody will buy the good quality e-cigarettes so…what's in that [e-cigarette vapour]? You obviously don't know, so I think in years to come they'll [bystanders] end up having problems. Might not be as bad as cigarette smoke, but I think the time to come…when they start doing research like that.

Consequently, there were some staff who expressed beliefs that e-cigarettes should be prohibited in Scottish prisons as a protective measure. Precautionary thinking about secondhand vapour (and the use of e-cigarettes) appeared to be influenced by several factors. As discussed, some staff appeared aware that there are gaps in scientific understanding of the health effects of e-cigarettes and secondhand vapour.

CC19 NS: I don't think there's enough research been done on the e-cigarettes to see if you can get passive smoking through that either so I wouldn't be happy with smokers

CC22 NS: So ban it until you've proved it's safe.

INTERVIEWER: So they're basically too new at the moment?

CC22 NS: Yes, that's it. You don't allow it until you can prove it's safe. This is what has not happened with tobacco.

Existing restrictions on vaping in many public places outside of the prison context appeared to reinforce perceptions that secondhand vapour might pose a danger to health:

NA06, S/ExV: It's the vapour itself. Which is why obviously it's been banned on trains and things like that, because they don't know enough about the vapour and the effect it might have on those around them. So you might be trading secondhand smoke for secondhand vapour. And it's…could be having exactly the same effect on people's lungs and everything else as the secondhand smoke does, so…

A desire to avoid past mistakes in relation to countries permitting the sale of cigarettes before the long-term health effects of smoking were known, perceptions that not enough is known about what is inside e-cigarettes and vapour, and beliefs that prisoners might use poor quality products are examples of further justifications given by staff for potentially prohibiting vaping in prisons.

#### Potential risks to prisoner health from vaping

Some staff worried about potential risks to prisoner health from the use of e-cigarettes, given that the long-term effects of vaping are not yet fully known. For example, one member of staff stated, 'I just don't think we should be giving them [prisoners] something [e-cigarettes] as a substitute that we don't even know a hundred per cent about.' (AA06 NS). Two other group members continued this discussion, saying:

AA07 ExS: Ten years down the line, we could all be smoking these vapes, and then we find out there's a risk.

AA01 NS : A brain tumour, or something.

Concerns about risks to prisoner health from vaping might also have been influenced by perceptions that e-cigarettes were being used, in wider society, as a long-term replacement for tobacco, rather than as a means to quit nicotine. For instance, one staff member (AA07, ExS)

said: 'I know loads of people that use them [e-cigarettes], and they say they're great. And you get addicted to that as well.' There were some suggestions that prisoners might struggle to give up vaping and that 'addiction' to e-cigarettes was undesirable. There was also a suggestion that it might be beneficial if prisoners had access to e-cigarettes on an interim (rather than permanent) basis to help them to manage cravings without forming a vaping habit. However, there were other staff who suggested that this might be unfair for prisoners who do not want to become abstinent and challenging for staff to manage.

> B01 NK: '…because again comparing it with the community, a lot of people do go for the e-cigarette, but like say they get addicted to it, and I just feel that we're trying to create a healthier lifestyle, and especially if they are confined, if they're allowed to smoke an e-cigarette, we're not helping them [prisoners], because they'll just keep doing it, and you'll find that they'll smoke more and more and more, so how's that helping them with a healthier lifestyle, whereas we're trying to get them totally off that.

### Potential risks to staff and prisoners from device misuse, product defects or accidents

As misuse of items was perceived to be an integral part of prison culture, there was a significant amount of discussion about potential ways in which prisoners might attempt to find alternative uses for e-cigarettes.

> NA06 S/ExV: Nothing's a hundred per cent tamper-proof and we're never going to find anything that's a hundred per cent tamper-proof…

Specific concerns focused on rechargeable devices and associated chargers and potential risk of devices being used to conceal contraband, 'smoke' illegal drugs (such as psychoactive substances) and charge prohibited mobile phones, for instance:

> EB12 NS: I mean, it's an alternative. And obviously, for your secondhand smoke, it's beneficial. But what could they do with these cartridges, and that sort of stuff? I mean, you're talking about people who can make anything out of anything…Is this just more stuff you're introducing to the jail, which they could use to do whatever?

There were also questions about the potential for devices to leak, explode or catch fire in prison. For example, one member of staff stated that e-cigarettes had received 'bad press' for 'blowing up in people's faces and people maybe have them in their pocket and it leaks or something and it burns' (CB01 NS).

### Perceptions of potential benefits of e-cigarettes in a smoke-free prison

While risks were identified, so were potential benefits of allowing e-cigarettes in a smoke-free prison. This included reducing harm to staff and prisoner health from smoking and SHS; helping prisoners to manage without tobacco, and the potential role of e-cigarettes in maintaining safety and discipline in prison, as described below.

### Reducing harm to staff and prisoner health from smoking and SHS

E-cigarettes were perceived by some staff to contain fewer harmful chemicals compared with conventional cigarettes and thus to be likely to pose fewer risks to the health of users:

> JA02 NS: I think they [e-cigarettes] are better for… smokers that want to come off smoking and they change from cigarettes to them because there are less carcinogens in but there is still something in them [e-cigarettes]. It has to be to create the vapour to carry the nicotine there's something in there.

Some staff also believed that secondhand vapour might be less dangerous and unpleasant for bystanders, with one staff member even saying the smell produced by some e-liquids was 'quite nice' (FA02, NK). Consequently, there was a suggestion that e-cigarettes might be beneficial for the health of everyone working and living in prisons when weighed against the dangers of smoking and SHS:

> FA02 NK: They could get the vapour thing, I don't know a lot about it, but vaping is safer than smoking. Why not let them vapour when they're in prison.
>
> FA05 NS: Im not being funny…
>
> FA02 NK: Better for us as well.
>
> FA05 NS: …I would rather somebody was vaping in the jail, than smoking some of the crap they smoke.
>
> FA02 NK: Yeah.
>
> FA03 NS: And it's got to be better for the smoker

### Helping prisoners to manage without tobacco

Some suggested that e-cigarettes could play a role in helping prisoners (including individuals who intended to smoke on their release from prison) to manage without tobacco should smoking be banned in prisons in the future. There was some discussion about the extent to which allowing prisoners to vape in prison was consistent with current practice in respect of treatment of prisoners addicted to other substances, such as illegal drugs like heroin.

> IA02 V: I totally agree [about the introduction of e-cigarettes into prisons]. I think it's been borne out with other…with the way we've treated other addictions, mainly methadone, I think there does need to be a substitute, it's by all accounts…the research thus far says it's a far, far cleaner substitute…

Lack of availability of 'medical' e-cigarettes in the UK, and uncertainties about the health effects of vaping were highlighted as points of difference between e-cigarettes and other 'substitute' products. Additionally, some implied that existing nicotine replacement products could fulfil a similar role to e-cigarettes.

### The potential role of e-cigarettes in maintaining safety and discipline in prison

There were some perceptions that, in the absence of tobacco, nicotine substitutes such as e-cigarettes and nicotine replacement therapy might help staff with the management and care of prisoners who were unable to smoke, especially new arrivals into custody:

> GA04 NS: I think if I was offering some…prisoners, you know, there's your cigarette, I'm taking them off you, there's a lollypop or there's an e-cigarette I think I'd rather give them an e-cigarette.

However, some expressed the view that it might be unfair if prisoners were permitted to vape in the future, since staff are not allowed to vape at work.

> CA14 NS: I just think, I don't know nothing about them either, but then if staff aren't allowed to bring them in.
>
> CA15 Ex: Why should prisoners be allowed?

Nicotine substitutes, alongside other measures, were also believed to have the potential to reduce organisational problems (eg, incidents of indiscipline, threats to staff safety and operational stability) associated with prison smoking bans. However, there was some discussion about whether and how substitutes for smoking might make the imposition of a smoking ban more achievable and help to diffuse challenging situations in smoke-free prisons.

> KA02 NS: Well, I think if it's [a smoking ban] managed properly and an alternative [to tobacco] is offered, you know, whether it's a certain e-cigarette or patches or something, then yes, it [a smoking ban] could work, but I think to have an outright ban with no alternative in place would just cause a hell of a problem.

## DISCUSSION

To our knowledge, this is the first study in any country to investigate staff views on the potential benefits and risks of introducing e-cigarettes for prisoners collected across an entire prison system before a decision to implement smoke-free policy. This evidence can assist with the development of acceptable and effective measures in respect of e-cigarettes in prisons and, in turn, support successful implementation of smoking bans and reduce tobacco-related harms, not just in Scotland but in other jurisdictions that are considering introducing smoke-free policy in the future.

We found evidence of discord among prison staff in Scotland about the overall balance of potential benefits and risks of e-cigarettes in a smoke-free prison; it is important to bear in mind that the data were collected before plans for a prison smoking ban were announced, and before e-cigarettes were available for purchase. We found that concerns about as yet unknown potential risks to health from e-cigarettes and secondhand vapour led some staff to feel apprehensive about the prospect of prisoners vaping. It is understandable that some prison staff showed precautionary attitudes about the possibility of replacing one workplace hazard (SHS exposures) with another that they thought could potentially be hazardous (prisoner vaping). The likelihood that some prisoners would try to modify devices, thus possibly causing harm to themselves or others, was cited as another potential risk of e-cigarettes in prisons. Significant discussion among staff about potential e-cigarette misuse or accidents might be, at least partly, explained by prison officers' primary responsibilities for maintaining operational safety and concerns about use of psychoactive substances in UK prisons.[22]

By contrast, staff support for allowing e-cigarettes in a smoke-free prison could be interpreted with reference to the principle of harm reduction. While e-cigarettes were believed to carry some risk, these risks were thought by some staff to be smaller than the certain dangers of smoking and SHS and the potential adverse consequences of removing tobacco from prisons. Given ongoing challenges in respect of supporting individuals to abstain from drug and alcohol use in prison, it is understandable that some staff believed that a range of nicotine substitutes should be offered in a smoke-free prison to help in the management of smoking addiction.

The finding that some within the staff group had misgivings and questions about e-cigarettes in smoke-free prisons in 2016/2017 was also reflected in a TIPs online survey of prison staff conducted at a similar time. The staff survey showed that 74% of staff (strongly) agreed that 'prison smoking bans are a good idea'. The proportion who (strongly) agreed that 'prison smoking bans are ok if prisoners are allowed e-cigarettes or vapes' was 36%. The equivalent TIPs survey of prisoners, conducted in the same time period, found evidence of stronger support for e-cigarettes among prisoners: while only 22% of prisoners (strongly) agreed that 'prison smoking bans are a good idea', prisoners expressed greater acceptance of bans (48%) if e-cigarettes were made available.[20]

A key strength, and novel element, of this paper is that it is based on analysis of rich qualitative data collected from a relatively large sample of prison staff with diverse smoking histories and experiences of working in varied prison settings and with different groups of prisoners. We therefore believe that our results provide a good indication of staff perspectives on the key potential benefits and risks of e-cigarettes in a smoke-free prison. We were able to collect such comprehensive data from Scottish prisons through close partnership working with senior staff with a remit for health and well-being, and others (such as representatives of employee Unions), in the Scottish Prison Service, starting with discussion of research plans in the pre-grant period. In line with the study design,[23] the research helped to inform and verify implementation strategies for smoke-free prisons in Scotland by feeding

back emergent findings from TIPs at monthly meetings of key stakeholders, including these findings on staff views on e-cigarettes, as prisons prepared to go smoke-free. We believe that the findings may be relevant to prisons in other countries who have adopted, or are considering, similar approaches to the UK on the regulation of tobacco and e-cigarettes.

Our study has four key limitations. First, focus group participants were self-selecting; it is notable that very few participating staff reported current/former vaping experience. A lower rate of current use (~3%) of e-cigarettes among staff focus group participants compared with the general population in Scotland (7%)[24] is perhaps not surprising given that prison staff are not allowed to use e-cigarettes in their place of employment. Second, there are gaps in the information about participants, notably the number of years of experience of working in prisons, which could have provided additional useful context to the results. Third, in some focus groups there were individuals who expressed strong views about whether to allow e-cigarettes in prisons. While TIPs researchers tried to ensure that diverse and opposing positions were captured during the process of collecting and analysing data, it is possible that some people may not have wanted to express their views in front of colleagues. As such, some positions may not be fully reflected in our findings due to group dynamics.[25] Finally, it is important to acknowledge that staff viewpoints on e-cigarettes may have developed since these qualitative data were collected.

The findings of our research suggest a number of measures which, taken together, might increase staff awareness and understanding of e-cigarettes, and enhance support for their use in smoke-free prisons. This is important, since e-cigarettes might be beneficial for the transition to and ongoing management of smoke-free policy; in particular, they may have the potential to enable prison staff to support strategies to increase prisoner motivation and capacity to achieve smoking abstinence, and ideally long-term cessation. By adding to the range of choices available to prisoners, e-cigarettes might help maximise the success, and health benefits, of smoke-free policy. Specifically, we suggest it would be beneficial if information on e-cigarettes were to be developed for prison settings which strikes a balance between taking a clear position on the relative harms of e-cigarettes compared with smoking tobacco or abstinence, while acknowledging limitations in the evidence. Such information might be valuable both for prison staff who are formally involved in health promotion work in prison (eg, Physical Education Instructors), as well as for prison staff who might be willing to provide opportunistic information and support in relation to e-cigarettes (and smoking behaviour) to prisoners. The findings support the sale of 'tamper proof' e-cigarettes in prison to protect staff and prisoner health. Additionally, they suggest it might be beneficial if frontline staff were offered training to enable them to swiftly identify when e-cigarettes are being used in ways which may cause serious injury to the user or other

people, as well as opportunities to feedback to management about the implications of e-cigarettes for prison security. Rules on the indoor use of e-cigarettes in prison (as has happened in Scotland) might be prudent, given concerns among some staff about potential risks of exposure to e-cigarette vapour, and possible residual frustrations about the decision to partially exempt prisons from national smoke-free laws when they were introduced in 2006.

Future research conducted after e-cigarettes and smoke-free policies have been introduced in Scottish prisons is needed to increase understanding of the real world implications of: allowing prisoners to buy e-cigarettes in smoke-free prisons; the ongoing management of people who enter prison as smokers; prison security; smoking cessation provision; and staff and prisoner attitudes and health. Subsequent phases of TIPs and a complementary study will provide evidence in respect of these questions.

In conclusion, our findings highlight that gaps in scientific evidence on e-cigarettes, misunderstanding about vaping, the complexity of prisons as workplaces and the distinctive nature of prison tobacco control policy all have implications for staff perceptions of the risks and benefits of e-cigarettes in smoke-free prisons. Reliable information on e-cigarettes embedded in wider health promotion work in prison, sale of 'tamper proof' products and rules on vaping indoors might reduce staff concerns and so help in the successful implementation and long-term success of smoke-free prisons.

**Acknowledgements** We are grateful to the staff who took part in the focus groups, and the Governors in Charge and staff at all 15 Scottish prisons who assisted with the overall study and facilitated access. We gratefully acknowledge the contribution of our co-investigators in TIPs research team to the overall design of the study (Dr Kathleen Boyd, Dr Philip Conaglen, Dr Peter Craig, Douglas Eadie, Professor Alastair Leyland, Professor Jill Pell,). We would particularly like to thank Sarah Corbett, Linda Dorward, Ruth Parker and members of the SPS Research Advisory Group for their helpful input during the design and conduct of this research, and to members of the SPS Smoke-free Stakeholder Advisory Group for their feedback on early findings of the study.

**Contributors** KH, HS, SS, ED and LB developed the study, with input from colleagues in the TIPs research team and the SPS Research Advisory Group. KH conducted most of the focus groups, with input from HS, ED and GL. AB and KH conducted independent initial analyses of the data, and AB conducted the more detailed framework analysis. AB drafted the manuscript and all revisions. All authors contributed to interpretation of the data and reviewing of the final paper.

**Funding** The TIPs study was funded by the National Institute for Health Research Public Health Research Programme (project number 15/55/44). HS and ED are also funded by UK Medical Research Council (MC_UU_12017/12, MC_PC_13027 to ED) and Chief Scientist Office (SPHSU-12).

**Disclaimer** The views and opinions expressed are those of the authors and do not necessarily reflect those of the Public Health Programme, NIHR, NHS or the Department of Health.

**Competing interests** None declared.

**Patient consent for publication** Not required.

**Provenance and peer review** Not commissioned; externally peer reviewed.

**Data sharing statement** No additional data are available.

and indication of whether changes were made. See: https://creativecommons.org/licenses/by/4.0/.

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
