## [Reviewer comments · BMJ Open]

ARTICLE DETAILS

TITLE (PROVISIONAL)	Views of prison staff in Scotland on the potential benefits and risks of e-cigarettes in smoke-free prisons: a qualitative focus group study
AUTHORS	Brown, Ashley; Sweeting, Helen; Semple, Sean; Bauld, Linda; Demou, Evangelia; Logan, Greig; Hunt, Kate

VERSION 1 – REVIEW

REVIEWER	James Woodall Leeds Beckett University, UK
REVIEW RETURNED	11-Dec-2018

GENERAL COMMENTS	This is an extremely important paper which is well-executed and covers a large sample of staff. The paper is extremely timely and will add significantly to understanding around the creation of healthy prison settings. I do, however, have some minor points for clarification: 1. What was the rationale for the paired interviews? I am suspecting this was not part of the design and rather due to low participation in some sites? If this is the case, can it be stated. If not, can it be explained why these were added and what the methodological rationale was.2. I was interested about the focus group composition. How were they practically composed? Was there any conscious effort to have smoking and non-smoking groups or just a mix of staff.3. Who were the staff? Were they all prison officers (and indeed what is meant by this exactly) working on the wings or were some in managerial roles, administrative roles etc. Were these all 'experienced' staff or were some new in post?4. I was curious why the mode of the focus group composition number was reported. Could there be a supplementary table instead which outlines the focus group numbers and breakdown of participants/prisons (anonymously)?5. The analysis and findings don't seem to tease out the differences, if any, between the sample. Do smoking and non-smoking staff have differing views? Were smoking staff members as concerned about second-hand vapour as non-smoking staff, for example? Were there any thematic nuances that can be reported?6. It would be helpful to make more explicit the similarities, distinctions and differences between this research and the complementary paper (reference 18) and how they interlink.
--

	7. Following on from the point above, there are five places where it says 'reported elsewhere' referring to reference 18. This becomes a little frustrating and maybe not always necessary. 8. I was unsure what the practice or policy suggestions are which come from this paper. In short, it suggests that more awareness and education of staff is needed. I wondered if this could be extended any further so that the paper has wider practical and political utility beyond SPS.
--	--

REVIEWER	Cheneal Puljevic Centre for Health Services Research, The University of Queensland, Australia
REVIEW RETURNED	12-Dec-2018

GENERAL COMMENTS	This manuscript describes a very important and under-researched topic: the use of e-cigarettes in prisons. With high levels of smoking recorded among people entering and leaving prisons, the provision of e-cigarettes in smoke-free prisons represent an important means of reducing smoking-related harm among an unhealthy population. Overall the paper is clearly written and structured. However, the main point that is lacking in my opinion is a justification for why this study is important, i.e. why are staff views about e-cigarettes important? This is hinted at in the key strengths and limitations section (i.e. these staff are “responsible for the management and care of vulnerable and/or challenging individuals subject to enforced smoking abstinence”) but I cannot see this explicitly mentioned anywhere in the paper. I recommend a few lines to the Introduction and/or Discussion making it clear why these staff members’ views are important within this context. Similarly, I think it is very important to more clearly discuss the potential implications of these staff views, especially in the Discussion, i.e. if a staff member continues to believe that vaping is very harmful, how will this affect their management of vaping devices within prisons? These staff members will be the ones responsible for distributing/ managing these vaping devices, and misinformation could potentially affect their willingness to support prisoners’ use of these devices. Secondly, although I know that this study follows a previously-published paper, I don’t believe that it is clear enough how survey data are relevant to this paper, and how the qualitative data fit within these survey data, i.e. are these participants a subset of the survey participants? I think that readers of this paper will benefit from increased understanding of how the two data sources complement each other. Similarly, I feel that the Discussion section could benefit from further in-depth explanation as to the implications/importance of the findings; at times findings are repeated but their implications/ explanations are not sufficiently described. For example, on page 17- “ We suggest it would be beneficial if specific information were to be developed for prison settings...”—further explanation is needed for *why* this would be beneficial. A few other small issues are listed below. Methods
--

	1. Bottom of page 7/ top of page 8: sentence “the researchers used the guide, formulating questions using their own words...” This sentence is a bit clumsy; I think an additional conjunction is needed. 2. Data collection: Where were focus groups conducted? How were people allocated to groups? 3. Page 8: Phase 1 survey data is mentioned; it is unclear at this stage how this data is relevant to this study. Further explanation needed. 4. Data analysis: The section would benefit from a little more detail (only 1-2 extra sentences) on how coding and thematic analyses were conducted. For example I feel like a bit more detail is needed about the “descriptive coding scheme” and how it was developed and applied, and how themes were developed and verified etc. This section is a little bare. Results 5. It would be potentially useful to understand more about the participants’ characteristics, especially the diversity of job roles and years of experience, and whether these may influence perceptions. This is alluded to in the Discussion e.g. “experiences of working in varied prison settings and with different groups of prisoners” so it may be useful to understand this varied experience in a little more detail. 6. Page 9: “Consequently, a notable feature across the group discussions was expressed low levels of knowledge”— please revise this sentence. 7. Page 12: spelling error in the quote: ‘affect’ should be ‘effect’ 8. Page 12: “A desire to avoid past mistakes which had been made in tobacco control”— it is not explicitly clear what is meant here 9. Bottom of page 12: “Concerns about risks to prisoner health from vaping might also have been influenced by perceptions that e-cigarettes were being used, in wider society, as a long-term replacement for tobacco, rather than as a means to cut down and then quit nicotine.”—are you able to add any quotes/ references to corroborate this assumption? 10. Page 13: “make weapons etc”—I feel that this sentence’s phrasing is a bit too colloquial 11. Page 16: “Nicotine substitutes, alongside other measures, were also believed to have the potential to reduce potential organisational problems”— this sentence could benefit from a revision (double use of the word potential)
--	---

REVIEWER	Stephanie Dugdale Breaking Free Group, England
REVIEW RETURNED	11-Jan-2019

GENERAL COMMENTS	This article reports on the perceptions of prison staff in Scotland around offender use of e-cigarettes. Data were reported before the smoking ban was implemented in Scotland, and e-cigarettes were made available alongside this. I think the study is of interest and was pertinent before the ban was brought in. Prisons in England also had similar concerns before the ban and before it was certain whether e-cigarettes would become available. Generally I found this to be a clear and well written paper. I would be happy to review any re-submissions. After having recently read the other TIPs paper, it is interesting to see where this article aligns with it. However, there are a few points that need addressing:
--

	-Introduction, page 1, line 25: missing 'as' - "Over issues such as..." -Introduction page 1, line 37: In first instance, spell out 'WHO' -Introduction, page 2, line 15: In first instance, spell out SHS - Introduction, page 2, lines 19-20: It might be useful to spell out that smoking ban applies to both inside and outside areas. It might also be worth noting that the ban applies to closed rather than open prisons -Introduction, page 3, line 1: the 14 Scottish prisons, is the one you did not access the closed one? Again just to make this clear if so. -Sampling and recruitment: Where there any other demographic details? -Data collection: I'd remove the last paragraph on this around Phase 1. Just report the findings of this study and keep it fully qualitative as reported in your aims. This in mind, I would recommend also removing the last sentence from the section 'Data analysis', and the first paragraph from the 'results'. -Data analysis: It looks as if you conducted deductive thematic analysis, I'd mention this. -Data analysis: Please mention your process of thematic analysis Results: Section under title 'personal experience and expressed knowledge...'. Is this a background to the themes rather than a theme itself? If so, I'd change the title to reflect this e.g. 'background to themes: personal experience and expressed knowledge of e-cigarettes' -Results: I'd refer to risks and benefits as the themes and groups of evidence within these as sub-themes e.g. 'potential risks to staff health' as a sub-theme -Results, theme 'perceptions of potential benefits...', sub-theme 'reducing harm to staff and prisoner health...': The last sentence of this on designated areas, is there some evidence to accompany this? -Discussion: That you were able to get the entire Scottish prison system to participate is fantastic. The process by which you did this is of real interest. Myself and other researchers have frequently struggled gaining access to prisons, and this is well reported in the literature, so some further explanation would be of great benefit. I know you mention this is through a contact within the prison, but again, any more information here would be good. -Discussion, page 15, line 34: please could you provide a reference on group dynamics -Discussion: This section may benefit from a separate 'conclusions' section. -References, number 7: check page numbers
--	--

REVIEWER	Hannah Walsh King's College London, UK
REVIEW RETURNED	22-Jan-2019

GENERAL COMMENTS	This is a good quality study exploring a timely and relevant topic in a setting which is not often investigated but very important from a tobacco control perspective, thank you for the opportunity to review it. Please see below for suggestions to improve the manuscript.
--

	Title: As per SRQR guidelines suggested by BMJ Open for qualitative studies, include either qualitative approach or data collection method in manuscript title. 2. "Objective" of abstract: this reads as if you have explored staff views post introduction of e-cigarettes and smoke-free policy: suggest re-wording to clarify timing, i.e. views were explored in advance of implementation of policy and introduction of e-cigarettes. 4. Further elucidation of qualitative analysis method would strengthen this section, I note that the data was part of a larger qualitative study reported elsewhere. In "data analysis" section, clarify rationale for using Framework function within NVivo, whereas overall method appears to be thematic analysis. 5. Use of Research Advisory Group is a strength of this study. 11. Though the discussion is clear and relevant, to add to the significance of this study, it would be useful to explore the relevance of prison staff attitudes towards e-cigarettes further, for example what implications might this have for prisoners use of e-cigarettes as a smoking cessation aid, what implications might the staff attitudes found have for the successful implementation of smoke-free policies. If the attitudes shown by staff are similar to the general public, as you have mentioned in the discussion, how does this translate into a prison setting (i.e. prisoners may be more influenced by their immediate surroundings than general public views, and prison staff may play a part in that influence). Secondly, it would be useful to know what the implications are from these findings for policy development in more depth, how might e-cigarettes play a part in reducing smoking related harms for prisoners, and what further research is suggested by these findings, i.e. is a similar study post-implementation of smoke-free policy and/or introduction of e-cigarettes warranted, how does it relate to studies of prisoner's attitudes towards e-cigarettes if there are any. 13. Inclusion of a completed SRQR in supplementary materials would further demonstrate the rigour of this study, but I wouldn't consider this essential.
--	--

VERSION 1 – AUTHOR RESPONSE

Response to reviewers: ‘Views of prison staff in Scotland on the potential benefits and risks of e-cigarettes in smoke-free prisons: a qualitative focus group study’ (Journal reference: 2018-027799’)

Reviewer: 1

“This is an extremely important paper which is well-executed and covers a large sample of staff. The paper is extremely timely and will add significantly to understanding around the creation of healthy prison settings.”

We thank the reviewer for these positive comments.

R1.1 What was the rationale for the paired interviews? I am suspecting this was not part of the design and rather due to low participation in some sites? If this is the case, can it be stated. If not, can it be explained why these were added and what the methodological rationale was.

We have added the following sentence (lines 146-148) to explain the rationale for the paired interviews:

The reason for carrying out paired interviews on two occasions was that other prison staff who were due to participate in the focus group were unable to attend at short notice.

R1.2 I was interested about the focus group composition. How were they practically composed? Was there any conscious effort to have smoking and non-smoking groups or just a mix of staff/ R.1.3 Who were the staff? Were they all prison officers (and indeed what is meant by this exactly) working on the wings or were some in managerial roles, administrative roles etc. Were these all 'experienced' staff or were some new in post?

On reading the reviewer comments, we have noted several requests for clarifications about our study methods and have now expanded this section of the paper. In respect of R1's query about focus group composition, the following information has been added on lines 150-154:

To enable the research to explore the diversity of views on smoking in prisons and prison smoking bans within and between establishments it was explained that ideally we would like the focus groups to include both smoking and non-smoking staff in a range of work roles. While we had limited control over how focus groups were assembled by the prison point of contact, most displayed some diversity with respect to staff smoking status.

We have also added the following information to our Methods (lines and 158-164):

Although it was not possible to record information on staff job roles consistently, the majority of those who took part were Scottish Prison Service staff, while some worked in the prison for other agencies, such as the National Health Service. Scottish Prison Service staff were a mix of residential, operational and instructor officers, managerial roles and administrative posts. The 14 prisons in which staff were working were varied with respect to: prisoner population (e.g. sex, age, and sentence length), capacity, security status, and prison architecture.

Unfortunately, we are unable to provide information in respect of years of experience in the role, as not all staff provided this information. We have added a sentence on this to our study limitations in the Discussion (lines 400-402) as follows:

Second, there are gaps in the information about participants, notably the number of years of experience of working in prisons, which could have provided additional useful context to the results.

R.1.4 I was curious why the mode of the focus group composition number was reported. Could there be a supplementary table instead which outlines the focus group numbers and breakdown of participants/prisons (anonymously)?

We have removed reference to the mode of the focus group composition number, but kept in information about the range of focus group sizes. We have revised the text in the Methods to provide more detailed information on the sample and group composition. We would be willing to provide a supplementary table which outlines the focus group numbers and participant breakdown (anonymously) if the editor thought this was essential.

R.1.5 The analysis and findings don't seem to tease out the differences, if any, between the sample. Do smoking and non-smoking staff have differing views? Were smoking staff members as concerned about second-hand vapour as non-smoking staff, for example? Were there any thematic nuances that can be reported?

We have revised the text to highlight the distinct views of staff who had some direct experience of vaping outside of work (lines 237-241):

Although only eight participants had experience of vaping, those with direct experience of e-cigarettes generally acknowledged their potential benefits in smoke-free prisons. However, there were

exceptions; for instance, less positive views were expressed by a staff member who had themselves stopped vaping due to concerns about potential adverse health effects of e-cigarette vapour.

We have clarified that concerns about e-cigarette vapour were expressed both by smoking and non-smoking staff on line 249.

R.1.6 It would be helpful to make more explicit the similarities, distinctions and differences between this research and the complementary paper (reference 18) and how they interlink.

We agree. We have re-ordered and edited some of the text so that this point is now explicitly addressed in the Introduction section of the paper on lines 117-125. The relevant sentences now read:

This paper extends previous reporting of prison staff and prisoner views on prison smoking bans, which only includes brief mention of the potential place of e-cigarettes in smoke-free prisons. Here, we use the staff focus group data to explore in detail staff views on the specific benefits and risks of e-cigarettes. The research could help with the development of strategies in respect of e-cigarettes in prison and so support the successful introduction of smoke-free policies, and help reduce tobacco-related harms, not just in Scotland (where prisons have recently gone smoke-free) but in other jurisdictions that are considering implementing bans in the future.

R1.7 Following on from the point above, there are five places where it says 'reported elsewhere' referring to reference 18. This becomes a little frustrating and maybe not always necessary.

On rereading the manuscript, we agreed that some of these references to our previous paper on staff and prisoner views on prison smoking bans were unnecessary and these have been cut.

R.1.8 I was unsure what the practice or policy suggestions are which come from this paper. In short, it suggests that more awareness and education of staff is needed. I wondered if this could be extended any further so that the paper has wider practical and political utility beyond SPS.

The other reviewers also very helpfully picked up this point too and asked that we state the implications of our results for policy and practice more explicitly. Hence, we have added a new paragraph to the Discussion (lines 410-432):

The findings of our research suggest a number of measures which, taken together, might increase staff awareness and understanding of e-cigarettes, and enhance support for their use in smoke-free prisons. This is important, since e-cigarettes might be beneficial for the transition to and ongoing management of smoke-free policy; in particular, they have the potential to enable prison staff to support strategies to increase prisoner motivation and capacity to achieve smoking abstinence, and ideally long-term cessation. By adding to the range of choices available to prisoners, e-cigarettes might help maximise the success, and health benefits, of smoke-free policy. Specifically, we suggest it would be beneficial if information on e-cigarettes were to be developed for prison settings which strikes a balance between taking a clear position on the relative harms of e-cigarettes compared to smoking tobacco or abstinence, while acknowledging limitations in the evidence. Such information might be valuable both for prison staff who are formally involved in health promotion work in prison (e.g. Physical Education Instructors), as well as for prison staff who might be willing to provide opportunistic information and support in relation to e-cigarettes (and smoking behaviour) to prisoners. The findings support the sale of 'tamper proof' e-cigarettes in prison to protect staff and prisoner health. Additionally, they suggest it might be beneficial if frontline staff were offered training to enable them to swiftly identify when e-cigarettes are being used in ways which may cause serious injury to the user or other people, as well as opportunities to feedback to management about the implications of e-cigarettes for prison security. Rules on the indoor use of e-cigarettes in prison (as has happened in Scotland) might be prudent, given concerns among some staff about potential risks of exposure to e-cigarette vapour, and residual frustrations about the decision to partially exempt prisons from national smoke-free laws when they were introduced in 2006.

We have also added a separate conclusion section to the Discussion on lines 439-445)

In conclusion, our findings highlight that gaps in scientific evidence on e-cigarettes, misunderstanding about vaping, the complexity of prisons as workplaces and the distinctive nature of prison tobacco control policy all have implications for staff perceptions of the risks and benefits of e-cigarettes in smoke-free prisons. Reliable information on e-cigarettes embedded in wider health promotion work in prison, sale of 'tamper proof' products and rules on vaping indoors might reduce staff concerns and so help in the successful implementation and long-term success of smoke-free prisons.

Finally, we have revised the conclusion of the Abstract (lines 43-47):

Our findings highlight that scientific uncertainty, misunderstanding about vaping, the complexity of prisons as workplaces and prison tobacco control policy all have implications for staff perceptions of the potential place of e-cigarettes in smoke-free prisons. To alleviate staff concerns, there is a need for reliable information on e-cigarettes. Staff may also require reassurances on whether products are 'tamper proof', and rules about vaping indoors.

Reviewer: 2

This manuscript describes a very important and under-researched topic: the use of e-cigarettes in prisons. With high levels of smoking recorded among people entering and leaving prisons, the provision of e-cigarettes in smoke-free prisons represent an important means of reducing smoking-related harm among an unhealthy population.

We thank the reviewer for these positive comments.

R2.1 Overall the paper is clearly written and structured. However, the main point that is lacking .. is a justification for why this study is important, i.e. why are staff views about e-cigarettes important? This is hinted at in the key strengths and limitations section .. but I cannot see this explicitly mentioned anywhere in the paper. I recommend a few lines to the Introduction and/or Discussion making it clear why these staff members' views are important within this context. Similarly, I think it is very important to more clearly discuss the potential implications of these staff views, especially in the Discussion.. These staff members will be the ones responsible for distributing/ managing these vaping devices, and misinformation could potentially affect their willingness to support prisoners' use of these devices.

We agree and have made a number of revisions and additions to the paper, which we hope make clearer the value and implications of our study for policy and practice.

In the Introduction, we have added the following sentences on the importance/value of the research on lines 107-110 and 117-125:

In-depth qualitative research examining employees' views on vaping in particular settings is required to help with the development of acceptable and effective workplace policies and measures on e-cigarettes.

This paper extends previous reporting of prison staff and prisoner views on prison smoking bans, which only includes brief mention of the potential place of e-cigarettes in smoke-free prisons. Here, we use the staff focus group data to explore in detail staff views on the specific benefits and risks of e-cigarettes. The research could help with the development of strategies in respect of e-cigarettes in prison and so support the successful introduction of smoke-free policies, and help reduce tobacco-related harms, not just in Scotland (where prisons have recently gone smoke-free) but in other jurisdictions that are considering implementing bans in the future.

We have also added a paragraph to the Discussion (lines 410-432) on the potential implications of our results:

The findings of our research suggest a number of measures which, taken together, might increase staff awareness and understanding of e-cigarettes, and enhance support for their use in smoke-free prisons. This is important, since e-cigarettes might be beneficial for the transition to and ongoing management of smoke-free policy; in particular, they have the potential to enable prison staff to support strategies to increase prisoner motivation and capacity to achieve smoking abstinence, and ideally long-term cessation. By adding to the range of choices available to prisoners, e-cigarettes

might help maximise the success, and health benefits, of smoke-free policy. Specifically, we suggest it would be beneficial if information on e-cigarettes were to be developed for prison settings which strikes a balance between taking a clear position on the relative harms of e-cigarettes compared to smoking tobacco or abstinence, while acknowledging limitations in the evidence. Such information might be valuable both for prison staff who are formally involved in health promotion work in prison (e.g. Physical Education Instructors), as well as for prison staff who might be willing to provide opportunistic information and support in relation to e-cigarettes (and smoking behaviour) to prisoners. The findings support the sale of ‘tamper proof’ e-cigarettes in prison to protect staff and prisoner health. Additionally, they suggest it might be beneficial if frontline staff were offered training to enable them to swiftly identify when e-cigarettes are being used in ways which may cause serious injury to the user or other people, as well as opportunities to feedback to management about the implications of e-cigarettes for prison security. Rules on the indoor use of e-cigarettes in prison (as has happened in Scotland) might be prudent, given concerns among some staff about potential risks of exposure to e-cigarette vapour, and residual frustrations about the decision to partially exempt prisons from national smoke-free laws when they were introduced in 2006.

We have also added a separate conclusion section to the Discussion on lines 439-445:

In conclusion, our findings highlight that gaps in scientific evidence on e-cigarettes, misunderstanding about vaping, the complexity of prisons as workplaces and the distinctive nature of prison tobacco control policy all have implications for staff perceptions of the risks and benefits of e-cigarettes in smoke-free prisons. Reliable information on e-cigarettes embedded in wider health promotion work in prison, sale of ‘tamper proof’ products and rules on vaping indoors might reduce staff concerns and so help in the successful implementation and long-term success of smoke-free prisons.

Finally, we have revised the conclusion of the Abstract (lines 43-47):

Our findings highlight that scientific uncertainty, misunderstanding about vaping, the complexity of prisons as workplaces and prison tobacco control policy all have implications for staff perceptions of the potential place of e-cigarettes in smoke-free prisons. To alleviate staff concerns, there is a need for reliable information on e-cigarettes. Staff may also require reassurances on whether products are ‘tamper proof’, and rules about vaping indoors.

R2.2. Secondly, although I know that this study follows a previously-published paper, I don’t believe that it is clear enough how survey data are relevant to this paper, and how the qualitative data fit within these survey data

In line with the recommendation of Reviewer 3, we now only briefly mention the results of the staff online survey for context in the Discussion on lines 373-381, since the objective of this paper is to qualitatively explore staff perceptions of the potential benefits and risks of e-cigarettes in smoke-free prisons.

R2.3 Similarly, I feel that the Discussion section could benefit from further in-depth explanation as to the implications/importance of the findings.... For example, on page 17- “ We suggest it would be beneficial if specific information were to be developed for prison settings...”—further explanation is needed for *why* this would be beneficial.

We agree. As detailed above in response to R1.8, we have made a number of changes to the Discussion (lines 410-432, 439-445) to explain the potential value and implications of our results for policy and practice.

Methods

R.2. 4 Bottom of page 7/ top of page 8: sentence “the researchers used the guide, formulating questions using their own words...” This sentence is a bit clumsy; I think an additional conjunction is needed.

We apologise for the clumsy wording. The sentence (lines 181-184) now reads:

The researchers formulated questions using their own words (often in response to issues raised by the groups), adjusted the order of topics as appropriate, prompted further discussion where relevant and invited staff to raise any points which they thought were pertinent.

R2.5 Data collection: Where were focus groups conducted? How were people allocated to groups?

This information is now included in the Methods on lines 174 and 148-154. We say:

They were carried out in a room in each prison chosen by the point of contact.

We asked the point of contact to invite around eight prison staff to participate in a focus group with other staff from the same prison. To enable the research to explore the diversity of views on smoking in prisons and prison smoking bans within and between establishments it was explained that ideally we would like the focus groups to include both smoking and non-smoking staff in a range of work roles. While we had limited control over how focus groups were assembled by the prison point of contact, most displayed some diversity with respect to staff smoking status.

R2.6 Page 8: Phase 1 survey data is mentioned; it is unclear at this stage how this data is relevant to this study. Further explanation needed.

As discussed above, we have removed mention of the results of the staff online survey from p8, since the objective of this paper is to qualitatively explore staff perceptions of the potential benefits and risks of e-cigarettes in smoke-free prisons.

R2.7 Data analysis: The section would benefit from a little more detail (only 1-2 extra sentences) on how coding and thematic analyses were conducted. For example I feel like a bit more detail is needed about the “descriptive coding scheme” and how it was developed and applied, and how themes were developed and verified etc. This section is a little bare.

We agree that it would be useful to provide a more detailed explanation of our approach to analysis. We have edited and expanded the ‘analysis’ section (lines 188-201) and it now reads:

With written consent from participants, focus groups were audio recorded and transcribed verbatim. Transcripts were checked and de-identified prior to management of the data. TIPs researchers (KH, HS, ED and GL) who conducted the fieldwork developed a descriptive coding scheme to bring together data on similar topics in preparation for detailed analysis. This coding scheme was devised using a combination of inductive and deductive techniques. The task of coding transcripts was split between TIPs researchers. Due to the relatively large volume of qualitative data, summaries with digital links to the raw data for all content relating to e-cigarettes were subsequently produced by AB using the Framework function in NVivo software (QSR international). AB used the data summaries and raw data to conduct thematic analysis. The process involved identifying different dimensions of staff opinions on e-cigarettes, grouping together dimensions which were similar to create themes and sub-themes and naming the themes and sub-themes. KH conducted independent analysis of the data, and other authors read a sample of the data to familiarise themselves. Emergent themes were discussed and revised until an interpretation was agreed on by all authors.

R2.8 It would be potentially useful to understand more about the participants’ characteristics, especially the diversity of job roles and years of experience, and whether these may influence perceptions. ...

We agree and have now provided additional information on the characteristics of the sample in the Methods on lines 150-164:

To enable the research to explore the diversity of views on smoking in prisons and prison smoking bans within and between establishments it was explained that ideally we would like the focus groups to include both smoking and non-smoking staff in a range of work roles. While we had limited control over how focus groups were assembled by the prison point of contact, most displayed some diversity with respect to staff smoking status. Across the sample of 132 staff, 78 had never smoked (NS); 30 were ex-smokers (ExS), and 11 currently used tobacco cigarettes (S). The smoking status for 13 participants is not known (NK). Eight staff reported having ever used an e-cigarette: five were currently vaping (V) and three were no longer vaping (ExV). Although it was not possible to record information on staff job roles consistently, the majority of those who took part were Scottish Prison Service staff, while some worked in the prison for other agencies, such as the National Health Service. Scottish Prison Service staff were a mix of residential, operational and instructor officers, managerial roles and administrative posts. The 14 prisons in which staff were working were varied with respect to: prisoner population (e.g. sex, age, and sentence length), capacity, security status, and prison architecture.

As noted under R1.3 above, we are unable to provide information in respect of years of experience in the role, since there is missing data and have added a sentence on this to our study limitations as follows. We have noted this in our study limitations (lines 400-402).

R2.9 Page 9: “Consequently, a notable feature across the group discussions was expressed low levels of knowledge”— please revise this sentence.

The sentence (lines 206-208) now reads:

Consequently, a recurring theme in the focus groups was staff reporting low levels of knowledge about e-cigarettes, contributing to a sense of uncertainty and confusion about vaping.

R2.10 Page 12: spelling error in the quote: ‘affect’ should be ‘effect’

We apologise for the error. This has been corrected.

R2.11 Page 12: “A desire to avoid past mistakes which had been made in tobacco control”- it is not explicitly clear what is meant here.

We have amended this sentence to clarify this (lines 262-263). It now reads:

A desire to avoid past mistakes in relation to countries permitting the sale of cigarettes before the long-term health effects of smoking were known....

R2.12 Bottom of page 12: “Concerns about risks to prisoner health from vaping might also have been influenced by perceptions that e-cigarettes were being used, in wider society, as a long-term replacement for tobacco, rather than as a means to cut down and then quit nicotine.”—are you able to add any quotes/ references to corroborate this assumption?

We have now added a quotation to illustrate the finding after line 285/p13.

R2.13 Page 13: “make weapons etc”—I feel that this sentence’s phrasing is a bit too colloquial

The sentence (line 292-294) now reads:

Specific concerns focussed on rechargeable devices and associated chargers and potential risk of devices being used to conceal contraband, ‘smoke’ illegal drugs (such as psychoactive substances) and charge prohibited mobile phones, for instance.

R2.14 Page 16: “Nicotine substitutes, alongside other measures, were also believed to have the potential to reduce potential organisational problems”- this sentence could benefit from a revision (double use of the word potential)

The wording has been revised.

Reviewer: 3

This article reports on the perceptions of prison staff in Scotland around offender use of e-cigarettes. Data were reported before the smoking ban was implemented in Scotland, and e-cigarettes were made available alongside this. I think the study is of interest and was pertinent before the ban was brought in. Prisons in England also had similar concerns before the ban and before it was certain whether e-cigarettes would become available.

Generally I found this to be a clear and well written paper.

We thank the reviewer for these positive comments.

R3.1 Introduction, page 1, line 25: missing 'as' - "Over issues such as..."

We apologise for the error. It has now been fixed.

R3.2 Introduction page 1, line 37: In first instance, spell out 'WHO'

We have done this.

R3.3. Introduction, page 2, line 15: In first instance, spell out SHS.

We have done this.

R3.4 Introduction, page 2, lines 19-20: It might be useful to spell out that smoking ban applies to both inside and outside areas. It might also be worth noting that the ban applies to closed rather than open prisons -Introduction, page 3, line 1: the 14 Scottish prisons, is the one you did not access the closed one? Again just to make this clear if so.

We now explicitly say that smoking is banned in all indoor and outdoor areas of prisons in England, Wales and Scotland.

In Scotland, all prisoners are subject to smoke-free rules while on prison grounds and in prison buildings, including those living in the open prison. We were only able to conduct focus groups in 14 out of the 15 prisons on this occasion because of difficulties arranging the focus group in the 15th prison within the available timescales; we were able to conduct a group within the open prison.

R3.5 Sampling and recruitment: Where there any other demographic details?

As noted in response to points R1.3 and R2.8, we have provided additional information on the characteristics of the sample in the Methods section (lines 150-164).

R3.6 Data collection: I'd remove the last paragraph on this around Phase 1. Just report the findings of this study and keep it fully qualitative as reported in your aims. This in mind, I would recommend also removing the last sentence from the section 'Data analysis', and the first paragraph from the 'results'.

We agree and we have now edited the text in line with this helpful suggestion.

R3.7 Data analysis: It looks as if you conducted deductive thematic analysis, I'd mention this/

R3.8 Data analysis: Please mention your process of thematic analysis

We have clarified that our analysis process involved a combination of inductive and deductive techniques and described our process of thematic analysis. The section on data analysis (lines 187-201) now reads:

TIPs researchers (KH, HS, ED and GL) who conducted the fieldwork developed a descriptive coding scheme to bring together data on similar topics in preparation for detailed analysis. This coding scheme was devised using a combination of inductive and deductive techniques. The task of coding transcripts was split between TIPs researchers. Due to the relatively large volume of qualitative data, summaries with digital links to the raw data for all content relating to e-cigarettes were subsequently produced by AB using the Framework function in NVivo software (QSR international). AB used the data summaries and raw data to conduct thematic analysis. The process involved identifying different dimensions of staff opinions on e-cigarettes, grouping together dimensions which were similar to create themes and sub-themes and naming the themes and sub-themes.¹⁹ KH conducted independent analysis of the data, and other authors read a sample of the data to familiarise themselves. Emergent themes were discussed and revised until an interpretation was agreed on by all authors.

R3.9 Results: Section under title 'personal experience and expressed knowledge...'. Is this a background to the themes rather than a theme itself? If so, I'd change the title to reflect this e.g. 'background to themes: personal experience and expressed knowledge of e-cigarettes'

Thank you for this suggestion. The section title now reads: 'Background: personal experience and expressed knowledge of e-cigarettes'.

R3.10 Results: I'd refer to risks and benefits as the themes and groups of evidence within these as sub-themes e.g. 'potential risks to staff health' as a sub-theme -Results, theme 'perceptions of potential benefits...', sub-theme 'reducing harm to staff and prisoner health...':

We agree and we now refer to themes and sub-themes in the manuscript.

R3. 11. The last sentence of this on designated areas, is there some evidence to accompany this?

We have now added a quote on p17 (line 336) to illustrate this finding.

R3.12 Discussion: That you were able to get the entire Scottish prison system to participate is fantastic. The process by which you did this is of real interest. Myself and other researchers have frequently struggled gaining access to prisons, and this is well reported in the literature, so some further explanation would be of great benefit. I know you mention this is through a contact within the prison, but again, any more information here would be good.

We have now explain in the Discussion (lines 386-393) that we were able to gain such comprehensive access to Scottish Prisons by working in close partnership with senior SPS staff with a remit for Health and Wellbeing in the pre-grant period. We plan to write a separate paper in the future to share our learning in respect of partnership working with SPS and in relation to the use of evidence to inform implementation of smoke-free policies in Scottish prisons. The text reads:

We were able to collect such comprehensive data from Scottish prisons through close partnership working with senior staff with a remit for health and wellbeing, and others (such as representatives of employee Unions), in the Scottish Prison Service, starting with discussion of research plans in the pre-grant period. In line with the study design,²¹ the research helped to inform and verify implementation strategies for smoke-free prisons in Scotland by feeding back emergent findings from TIPs at monthly meetings of key stakeholders, including these findings on staff views on e-cigarettes, as prisons prepared to go smoke-free.

R3.13 Discussion, page 15, line 34: please could you provide a reference on group dynamics

We have added a reference on group dynamics.

R3.13 Discussion: This section may benefit from a separate 'conclusions' section.

We agree and have added a separate 'conclusions' section (lines 439-445) which reads:

In conclusion, our findings highlight that gaps in scientific evidence on e-cigarettes, misunderstanding about vaping, the complexity of prisons as workplaces and the distinctive nature of prison tobacco control policy all have implications for staff perceptions of the risks and benefits of e-cigarettes in smoke-free prisons. Reliable information on e-cigarettes embedded in wider health promotion work in prison, sale of 'tamper proof' products and rules on vaping indoors might reduce staff concerns and so help in the successful implementation and long-term success of smoke-free prisons.

R3.14 References, number 7: check page numbers

The relevant journal advises that 'from the first issue of 2016, MDPI journals use article numbers instead of page numbers. See further details here.' However, we happy to take editorial advice on how to cite this article.

Reviewer: 4

This is a good quality study exploring a timely and relevant topic in a setting which is not often investigated but very important from a tobacco control perspective, thank you for the opportunity to review it.

We thank the reviewer for these positive comments.

R4.1 Title: As per SRQR guidelines suggested by BMJ Open for qualitative studies, include either qualitative approach or data collection method in manuscript title.

The study title has been changed to: 'Views of prison staff in Scotland on the potential benefits and risks of e-cigarettes in smoke-free prisons: a qualitative focus group study'.

R4.2 "Objective" of abstract: this reads as if you have explored staff views post introduction of e-cigarettes and smoke-free policy: suggest re-wording to clarify timing, i.e. views were explored in advance of implementation of policy and introduction of e-cigarettes.

Thank you for pointing this out. We have changed to text (lines 22-25) to read as follows:

Electronic cigarettes (e-cigarettes) were introduced into all Scottish prisons in February 2018, some months after prisons began preparing for a smoking ban in November 2018. In 2016/2017, prison staff views on the potential benefits and risks of e-cigarettes were explored in advance of the introduction of: (1) a smoking ban and (2) e-cigarettes.

R4.3 Further elucidation of qualitative analysis method would strengthen this section, I note that the data was part of a larger qualitative study reported elsewhere. In "data analysis" section, clarify rationale for using Framework function within NVivo, whereas overall method appears to be thematic analysis.

In amending the Methods section, we have clarified the rationale for using the Framework function within NVivo. Given the large volume of qualitative data, our approach to analysis involved two steps. First, the Framework Method was used to manage data in preparation for detailed analysis and interpretation. Second, data were thematically analysed working with data summaries and digital links to raw data which had been generated using the Framework Method. The text on lines 187-201 now reads:

TIPs researchers (KH, HS, ED and GL) who conducted the fieldwork developed a descriptive coding scheme to bring together data on similar topics in preparation for detailed analysis. This coding scheme was devised using a combination of inductive and deductive techniques. The task of coding transcripts was split between TIPs researchers. Due to the relatively large volume of qualitative data, summaries with digital links to the raw data for all content relating to e-cigarettes were subsequently produced by AB using the Framework function in NVivo software (QSR international). AB used the data summaries and raw data to conduct thematic analysis. The process involved identifying different dimensions of staff opinions on e-cigarettes, grouping together dimensions which were similar to create themes and sub-themes and naming the themes and sub-themes.

R4.4 Use of Research Advisory Group is a strength of this study.

Thank you.

R4.5 Though the discussion is clear and relevant, to add to the significance of this study, it would be useful to explore the relevance of prison staff attitudes towards e-cigarettes further, for example what implications might this have for prisoners use of e-cigarettes as a smoking cessation aid, what implications might the staff attitudes found have for the successful implementation of smoke-free policies. If the attitudes shown by staff are similar to the general public, as you have mentioned in the discussion, how does this translate into a prison setting (i.e. prisoners may be more influenced by their immediate surroundings than general public views, and prison staff may play a part in that influence). Secondly, it would be useful to know what the implications are from these findings for policy development in more depth, how might e-cigarettes play a part in reducing smoking related harms for prisoners, and what further research is suggested by these findings, i.e. is a similar study post-implementation of smoke-free policy and/or introduction of e-cigarettes warranted, how does it relate to studies of prisoner's attitudes towards e-cigarettes if there are any.

The other reviewers also very helpfully picked up this point and asked that we state the potential importance and implications of our results for policy and practice.

As notes in response to points to 1.8 and 2.3, we have made several changes to the Introduction (lines 107-109 and 117-125)) and Discussion (lines 410-432 and 439-445) to highlight the potential importance and implications of this study and its results.

We have added a new paragraph exploring areas for future research on lines 433-438:

Future research conducted after e-cigarettes and smoke-free policies have been introduced in Scottish prisons is needed to increase understanding of the real world implications of: allowing prisoners to buy e-cigarettes in smoke-free prisons; the ongoing management of people who enter prison as smokers; prison security; smoking cessation provision; and staff and prisoner attitudes and health. Subsequent phases of TIPs and a complementary study will provide evidence in respect of these questions.

We have also added a few sentences (lines 373-381) comparing the results of this study to our previous TIPs paper, since this is the only research we are aware of which explicitly examines staff and prisoner perceptions of e-cigarettes in smoke-free prisons.

The finding that some within the staff group had misgivings and questions about e-cigarettes in smoke-free prisons in 2016/2017 was also reflected in a TIPs online survey of prison staff conducted at a similar time. The staff survey showed that 74% of staff (strongly) agreed that 'prison smoking bans are a good idea'. The proportion who (strongly) agreed that 'prison smoking bans are ok if prisoners are allowed e-cigarettes or vapes' was 36%. The equivalent TIPs survey of prisoners, conducted in the same time period, found evidence of stronger support for e-cigarettes among prisoners: whilst only 22% of prisoners (strongly) agreed that 'prison smoking bans are a good idea', prisoners expressed greater acceptance of bans (48%) if e-cigarettes were made available.

R4.6 Inclusion of a completed SRQR in supplementary materials would further demonstrate the rigour of this study, but I wouldn't consider this essential.

Thank you for this suggestion. We agree and we have completed the SRQR checklist.

VERSION 2 – REVIEW

REVIEWER	Stephanie Dugdale Breaking Free Group, England
REVIEW RETURNED	25-Mar-2019

GENERAL COMMENTS	The authors addressed reviewer comments well, and I believe this paper to be suitable for publication. A few revisions to consider:  - Is there a reference for roll-out of smoking bans (page 5, lines 94-98)? - Results may be improved by adding a quote for staff who have stopped vaping due to health concerns (page 11, lines 241-243), and for perceptions of vaping as a replacement than smoking cessation tool (page 13, lines 279-281).
---

REVIEWER	Hannah Walsh King's College London, United Kingdom
REVIEW RETURNED	03-Apr-2019

GENERAL COMMENTS	Thank you for the opportunity to review this revised manuscript. The authors have made changes which strengthen the work considerably, including a comprehensive and detailed description of the methods used for data management and analysis, and an extended discussion section. I have made only a few points for re-wording to provide absolute clarity, hence minor revision recommended. I 23: suggested adding "before the implementation of the smoking ban in Nov..." in order to be crystal clear you are referring to the ban, not the preparation for the ban. Overall the descriptions of timings throughout are much clearer and easier to follow. I 235 suggest replacing "should" with "whether". I 251 "could worry" - unclear if they did or had the potential to worry - worth clarifying I 357 add in "before e-cigarettes were available for purchase" to make it absolutely clear.
---

VERSION 2 – AUTHOR RESPONSE

Reviewer: 3. The authors addressed reviewer comments well, and I believe this paper to be suitable for publication.

A few revisions to consider:

3.1 Is there a reference for roll-out of smoking bans (page 5, lines 94-98)?

We have done this.

3.2 Results may be improved by adding a quote for staff who have stopped vaping due to health concerns (page 11, lines 241-243), and for perceptions of vaping as a replacement than smoking cessation tool (page 13, lines 279-281).

We have added a quote for the point made on lines 279-281.

Reviewer: 4. Thank you for the opportunity to review this revised manuscript. The authors have made changes which strengthen the work considerably, including a comprehensive and detailed description of the methods used for data management and analysis, and an extended discussion section.

I have made only a few points for re-wording to provide absolute clarity, hence minor revision recommended.

4.1 | 23: suggested adding "before the implementation of the smoking ban in Nov..." in order to be crystal clear you are referring to the ban, not the preparation for the ban. Overall the descriptions of timings throughout are much clearer and easier to follow.

The sentence now reads: Electronic cigarettes (e-cigarettes) were introduced into all Scottish prisons in February 2018, some months after prisons began preparing in 2017 for a smoking ban implemented in November 2018.

4.2. | 235 suggest replacing "should" with "whether".

The sentence now reads: Hence, it was against this background that staff were evaluating the potential benefits and risks of vaping in prison, in the event that tobacco was removed from the prisons at some future date.

4.3 | 251 "could worry" - unclear if they did or had the potential to worry - worth clarifying

We agree that the sentence should be clarified and we have done this.

4.4. I 357 add in "before e-cigarettes were available for purchase" to make it absolutely clear.

We have done this.